# Overlapping upstream ORFs ending at c.125 lead to reduced Endoglin, contributing to Hereditary Hemorrhagic Telangiectasia
Carole Proust[1], Clémence Deiber[1], Caroline Meguerditchian[1], Maud Tusseau[2], Alexandre Guilhem[3,4,5], Shirine Mohamed[6], Aurélie Goyenvalle [7], Béatrice Jaspard-Vinassa[8], Sophie Dupuis-Girod[2,9], David-Alexandre Trégouët [1] ✉ & Omar Soukarieh [1,8] ✉

Hereditary Hemorrhagic Telangiectasia (HHT) is a rare vascular disease mainly caused by pathogenic mutations in *ACVRL1* and *ENG* genes. Despite advances in HHT diagnosis, the molecular origin of some cases remains unclear. Recently, we observed a high prevalence of HHT-causing 5'UTR variants in *ENG*. These variants commonly introduce upstream AUG codons (uAUGs) at the origin of upstream open reading frames (upORFs) overlapping the coding sequence, all terminating at the same stop codon located at position c.125 (uAUG-c.125). Here, we analyzed all 5'UTR *ENG* single nucleotide variants that could alter upORFs in silico. Interestingly, we found that 85% of uAUG-c.125 variants alter the protein levels. Furthermore, we identified 2 variants creating uAUG-c.125 and uCUG-c.125 in HHT patients and experimentally demonstrated their association with reduced endoglin levels This study provides new elements for the interpretation of upORF-altering variants in the 5'UTR of *ENG* with new insights for the molecular diagnosis of HHT.

Hereditary Hemorrhagic Telangiectasia (HHT) is a multiorgan rare vascular disease[1] caused by pathogenic mutations in *ACVRL1* and *ENG* genes in more than 90% of cases[2]. The clinical diagnosis of HHT is based on Curaçao criteria which include family history, epistaxis, multiple telangiectasias, and visceral vascular lesions, such as gastrointestinal telangiectasia and/or arteriovenous malformations (AVMs)[3]. Depending on the identified criteria in patients, the clinical diagnosis of HHT can be classified as definite, possible, or unlikely. Tremendous efforts have been made in order to identify new genetic drivers[2] and molecular explanations at the origin of HHT[4,5]. However, around 10% of cases are still with unclear molecular origin[1,2]. Unresolved cases can be attributed to rare variants in poorly explored and complex regions of the genome that have yet to be identified and characterized[6,7]. In this study, we aim to contribute to the

optimization of HHT molecular diagnosis related to the involvement of rare variants in HHT.

For a long time, the search for rare pathogenic mutations in HHT patients was mainly restricted to exonic/flanking intronic regions[1,5]. However, we and others have recently reported five rare pathogenic variations in the 5'UTR of *ENG* identified in HHT patients with definite diagnosis[8–13]. Acting as Loss-of-function mutations, these variations create upstream AUGs (uAUGs) in frame with the same stop codon in position c.125 (uAUG-c.125) and resulting in overlapping upstream Open Reading Frames (uoORFs). In addition, these variants were associated with a decrease in the protein levels of Endoglin, encoded by the *ENG* gene. Endoglin is a membrane glycoprotein that plays a major role in endothelial cell biology and is highly involved in angiogenesis[14,15]. It has mainly been

[1]Univ. Bordeaux, INSERM, Bordeaux Population Health Research Center, UMR 1219, Bordeaux, France. [2]Hospices Civils de Lyon, French National HHT Reference Center and Genetics department, Hôpital Femme-Mère-Enfant, Bron, France. [3]Hospices Civils de Lyon, Service de Génétique, Groupement Hospitalier Est, Bron, France. [4]Centre de Référence National pour la maladie de Rendu-Osler, Groupement Hospitalier Est, Bron, France. [5]TAI-IT Autoimmunité Unit RIGHT-UMR1098, Université de Bourgogne, INSERM, EFS-BFC, Besançon, France. [6]Département de Médecine interne et Immunologie Clinique, CHRU BRABOIS, Vandœuvre-lès-Nancy, France. [7]Université Paris-Saclay, UVSQ, Inserm, END-ICAP, Versailles, France. [8]Univ. Bordeaux, INSERM, Biology of Cardiovascular Diseases, U1034, Pessac, France. [9]Univ. Grenoble Alpes, Inserm, CEA, Laboratory Biology of Cancer and Infection, Grenoble, France. ✉e-mail: david-alexandre.tregouet@u-bordeaux.fr; omar.soukarieh@inserm.fr

known as a co-receptor in response to bone morphogenetic proteins belonging to the transforming growth factor $\beta$ superfamily[16–18].

Our previous data suggested that *ENG* could be enriched in HHT-causing uAUG-creating variants carrying the common characteristic to create overlapping upORFs ending with the same stop codon (c.125). Furthermore, given the growing evidence that non-canonical translation initiation sites (TIS) differing by one nucleotide from the AUG codon can also initiate translation[19,20], and despite such evidence in the context of HHT, we hypothesize that upstream ORFs (upORFs) starting with non-canonical TIS could also contribute to regulate the translation of the Endoglin coding sequence (CDS). Interestingly, upstream CUGs have particularly been described to be associated with human diseases[21,22]. To improve the molecular diagnosis of HHT, we here extracted all possible single nucleotide variants (SNVs) in the 5'UTR of *ENG* that could create or alter upORFs from the MORFEEdb database[23] resulting from the latest version of the bioinformatics tool MORFEE. In addition, we completed the functional analysis of all uAUG-c.125 variants and found that 85% of them are associated with a decrease in Endoglin levels in vitro. Finally, we identified a uAUG- and a uCUG- creating variants in French HHT patients and found

that they also decrease protein levels in our assays. This study will facilitate the detection and interpretation of non-coding variants in HHT, contributing to improving the molecular diagnosis for patients and their families.

## Results

### Identification of all possible upORF-altering variations in the 5'UTR of *ENG*

A total of 909 SNVs resulted from the in silico mutational saturation of the 5'UTR of *ENG* (Supplementary Data 1). MORFEE annotated 328 of them as creating uTIS, uStop, and/or deleting existing uStop. The annotated variations can be at the origin of 360 upORFs (Fig. 1a, Supplementary Data 2). More precisely, 255 variants are predicted to create new uTIS, 30 to create new uStop, and 12 to delete existing uStop (Fig. 1a). The remaining 31 SNVs show multiple consequences (Fig. 1a; Supplementary Data 3). While the majority of annotated variants do not exist in databases and correspond to artificial ones, 29 of them are reported in ClinVar as candidates for HHT. Of the 29 ClinVar *ENG* upORF variants, 27 are predicted to create uTIS (Supplementary Data 4), among which 21 are classified as variants of

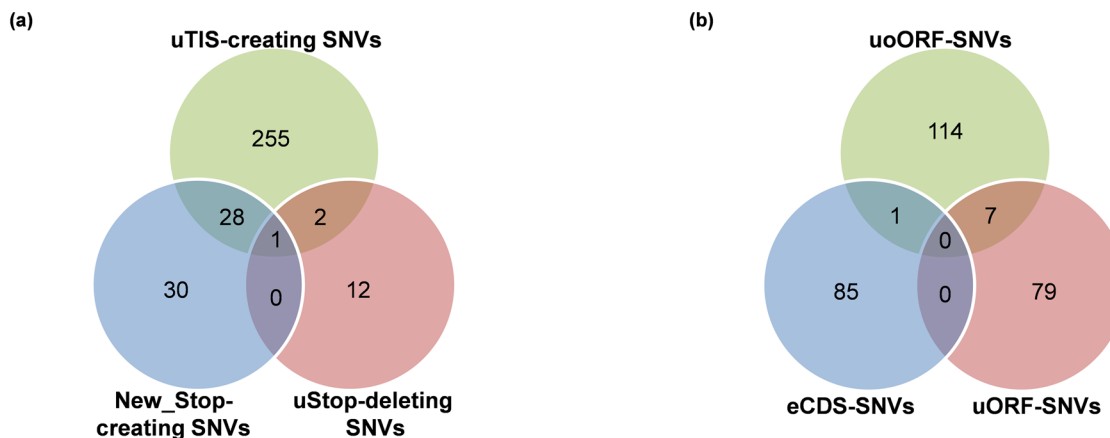

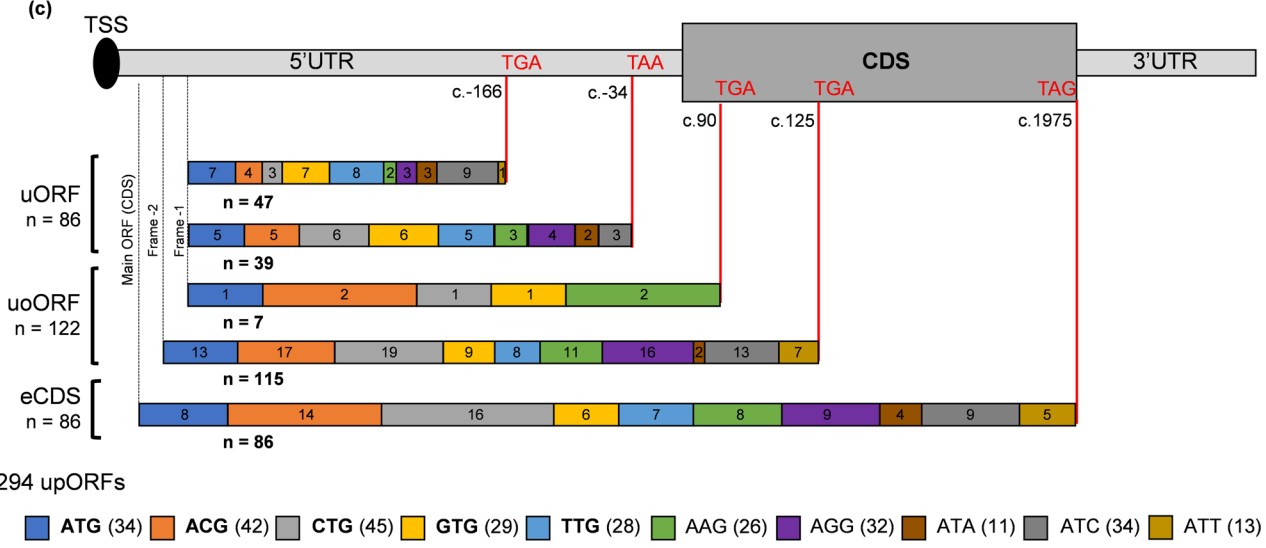

**Fig. 1 | Identification of all possible single nucleotide variants (SNVs) modifying or creating upstream open reading frames (upORFs) in the 5'UTR of *ENG*. a** MORFEE annotated SNVs that modify existing upORFs or create new ones (New_Stop-creating, uStop-deleting SNVs and uTIS-creating SNVs). **b** Three types of upORFs can result from the creation of upstream translation initiation sites (uTIS) in the 5'UTR of *ENG*. uoORF-SNVs, variants at the origin of upORFs overlapping the coding sequence (CDS); eCDS-SNVs, variants creating elongated CDS; uORF-SNVs, variants creating fully upstream upORFs. **c** Detailed illustration of upORFs generated by uTIS-creating SNVs. The type of uTIS and position of stop codons associated with the generated upORFs, as well as the number of each type of upORF and of uTIS are indicated. TSS, translation start site.

**Table 1 | Characteristics of the 13 uAUGs and the studied uCUG created by variants in the 5'UTR of *ENG***

| Variants | uTIS-SNV | Protein levels in vitro (%)‡ | uoORF size (nt) | Kozak sequence | TIS-Predictor (KSS score) | PreTIS score |
|---|---|---|---|---|---|---|
| Previously studied[13] | c.-142A > T$ | [0–14] | 270 | CAGATG**G** | **0.66** | **0.84** |
| | c.-127C > T#,$ | [0–5] | 255 | **G**GGATGC | **0.69** | **0.84** |
| | c.-79C > T#,$ | [8–27] | 207 | CCCATGC | **0.65** | **0.67** |
| | c.-68G > A | [30–46] | 195 | CCGATGC | 0.61 | **0.89** |
| | c.-10C > T#,$ | [21–49] | 138 | CCCATGT | 0.61 | **0.85** |
| New | c.-287C > A* | [61–115] | 414 | CATATGC | 0.42 | na |
| | c.-271G > T* | [4–19] | 399 | **G**CCATGA | **0.78** | na |
| | c.-249C > G | [6–12] | 378 | **G**CCATGC | **0.67** | na |
| | c.-182C > A | [4–17] | 309 | TCCATGT | 0.57 | **0.91** |
| | c.-167C > A | [7–77] | 294 | CTCATGA | 0.55 | **0.87** |
| | c.-37G > T* | [94–251] | 165 | CACATGA | 0.57 | **1** |
| | c.-33A > G | [0–25] | 162 | **A**GGATGA | **0.73** | **1** |
| | c.-31G > T | [12–41] | 159 | **A**TAATGC | 0,63 | **0.96** |
| | c.-76C > T#,$ | [50–182] | 204 | **A**CGCTG**G** | **0.87** | **0.93** |

uTIS (upstream Translation Initiation Start site) position in the L-*ENG* (Long isoform); NM_001114753.3 transcript (c.1 corresponds to the A of the main AUG) is indicated. Nucleotides at positions -3 and +4 relative to the predicted uTIS are in bold when corresponding to the most conserved nucleotide. The uTIS are underlined. Two additional scores have also been extracted from TIS-predictor[29] and PreTIS[30] tools, to predict the strength of Kozak sequence and the translation efficiency of the created uTIS, respectively. Bolded scores are those higher than the predefined thresholds.
$, the c.-142A > T, c.-127C > T, c.-79C > T, c.-10C > T, and c.-76C > T variants are reported in ClinVar database. The c.-142A > T, c.-127C > T are classified as pathogenic/likely pathogenic, the c.-79C > T and c.-76C > T as variants of unknown significance, and the c.-10C > T has conflicting interpretations, according to ClinVar. #, the c.-127C > T, c.-79C > T, c.-10C > T, and c.-76C > T variants are repoted in dbSNP database as rs1060501408, rs1564466502, rs756994701, and rs943786398, respectively. *, in addition of the uAUG-creation, c.-287C > A and c.-37G > T are predicted to create new stop codons shortening existing upORFs, and c.-271G > T is predicted to simultaneously create a uAUA at the origin of a fully upstream upORF. nt, nucleotide; ‡ Range of Endoglin protein levels observed across five independent experiments from Soukarieh et al.[13], and this study; na, non-applicable.

unknown significance (VUS) or with conflicting interpretations. Moreover, at least 14 *ENG* uTIS-creating variants have been reported in GnomAD V4 0.0 with minor allele frequency lower than 0.01% but without any evidence of association with HHT (Supplementary Data 5).

In more detail, 286 SNVs creating canonical or non-canonical uTIS could be at the origin of 294 upORFs, with the understanding that a given SNV may create different uTIS (Fig. 1b; Supplementary Data 6). Among these upORFs, 86 are fully located in the 5'UTR (uORFs) ending at two different stop codons (c.-166 and c.-34), 122 are overlapping with the CDS (uoORFs) ending with stop codons at position c.90 or c.125, and 86 correspond to elongated CDS ending at the main stop codon (i.e., natural stop codon of the CDS) (Fig. 1c).

Interestingly, 8 upORFs have been reported in public databases (www.sORFs.org[24]; https://metamorf.hb.univ-amu.fr/[25]; https://vutr.rarediseasegenomics.org/; and https://smorfs.ddnetbio.com[26]) as naturally existing in the 5'UTR of *ENG*. Five of these upORFs end with the stop codon located at c.-166, 2 with the stop codon at position c.-33 and 1 is overlapping the CDS and ends at the uStop-c.125 (Supplementary Fig. 1a). MORFEE annotated 8 variations as deleting the uStop codon located at position c.-166, and 7 that delete the uStop codon at position c.-34, thus elongating these existing upORFs into either longer fully uORFs ending at positions c.-34 or into uoORFs ending at new stop codon c.90, respectively (Supplementary Fig. 1b). By contrast, MORFEE identified 17 variations that could create new stop codons then shortening existing upORFs reported in databases and 42 that could create new stop codons at the origin of new upORFs (Supplementary Fig. 1c). Of note, two variants creating new stop codons (c.-200C > T and c.-133C > T) have been reported as VUS in ClinVar and two others (c.-253C > A and c.-87C > T) have been reported as rare in GnomAD V4 0.0 database with the c.-87C > T showing multiple consequences on upORFs (Supplementary Data 3&5).

### Most *ENG* SNVs creating uAUGs-c.125 drastically alter the protein levels

Recently, we have demonstrated that 5 *ENG* 5'UTR variants identified in HHT patients and creating uAUG-initiated uoORFs all ending at the uStop-

c.125, were responsible for decreased protein levels[13], revealing their pathogenicity (Table 1). To extrapolate whether these deleterious effects also hold for any other uAUG-c.125, we conducted the same experimental work[13] on the remaining 8 variations predicted by the MORFEE in silico analysis (Fig. 2a; Table 1). No decrease of protein levels was observed for the c.-287C > A and c.-37G > T variations (Fig. 2b, c; Supplementary Fig. 2; Supplementary Data 7). While the c.-287C > A variant showed similar protein levels comparing to the wild-type (WT), the c.-37G > T tended to associate with an increase of Endoglin levels in our assay (Fig. 2b-c; Supplementary Fig. 3a). Interestingly, the c.-37G > T variant is predicted to simultaneously create a uAUG-initiated uoORF and to shorten an existing upORF (Supplementary Data 3). The six remaining variations were associated with a mean protein level lower than 40% compared to the WT construct. After correcting for multiple testing, the association was statistically significant ($p < 0.05$) for four of these variants (c.-271G > T, c.-249C > G, c.-128C > A, c.-33A > G). Despite similar trends, the other two (c.-167C > A and c.-31G > T) did not reach significance ($p \sim 0.10$). No significant difference was observed in our RT-qPCR experiments between mutants and WT (Supplementary Fig 3b). In total, 85% (11/13) of these variants (current and previous works[13]) decrease the Endoglin protein levels in vitro.

Among the 13 5'UTR variants we evaluated in vitro, six are characterized by a moderate Kozak sequence (Table 1). These six variants (c.-271G > T, c.-249C > G, c.-142A > T, c.-127C > T, c.-33A > G and c.-31G > T) were all associated with decreased protein levels in our assay with 5/6 variants showing the most drastic effects. The seven remaining variants were associated with a weak Kozak sequence and five of them (c.-182C > A, c.-167C > A, c.-79C > T, c.-68G > A and c.-10C > T) were associated with a decrease of the protein levels. Finally, the two variants with no decrease of Endoglin protein levels (c.-287C > A and c.-37G > T) are associated with weak Kozak sequences. Our analysis suggests that the sole information on the Kozak sequence of uTIS cannot be used to predict the potential effect of uAUG-creating variants in *ENG*.

We extended the predictions of the Kozak sequence by applying predictions from TIS-predictor, which takes into account the ten nucleotides

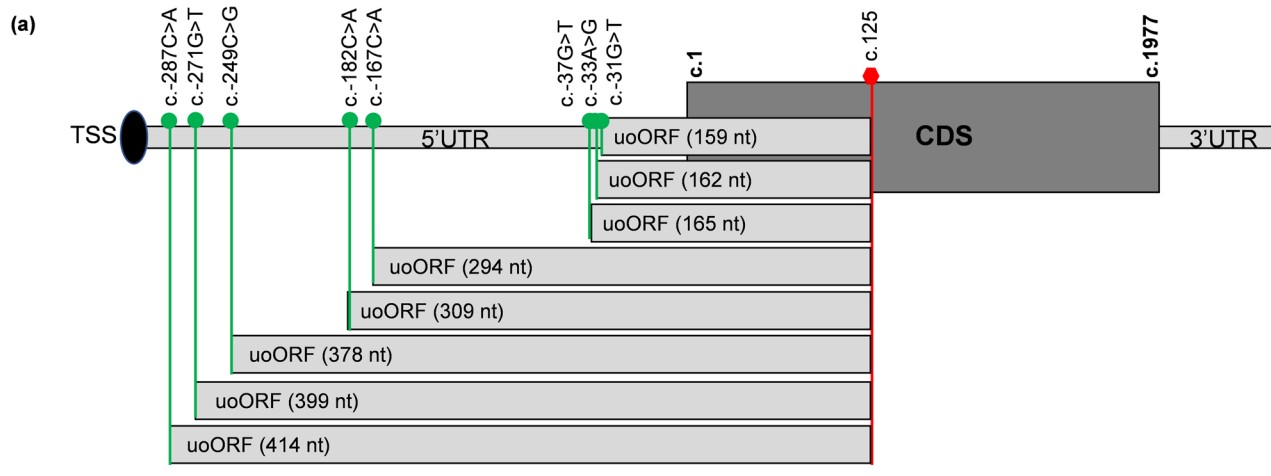

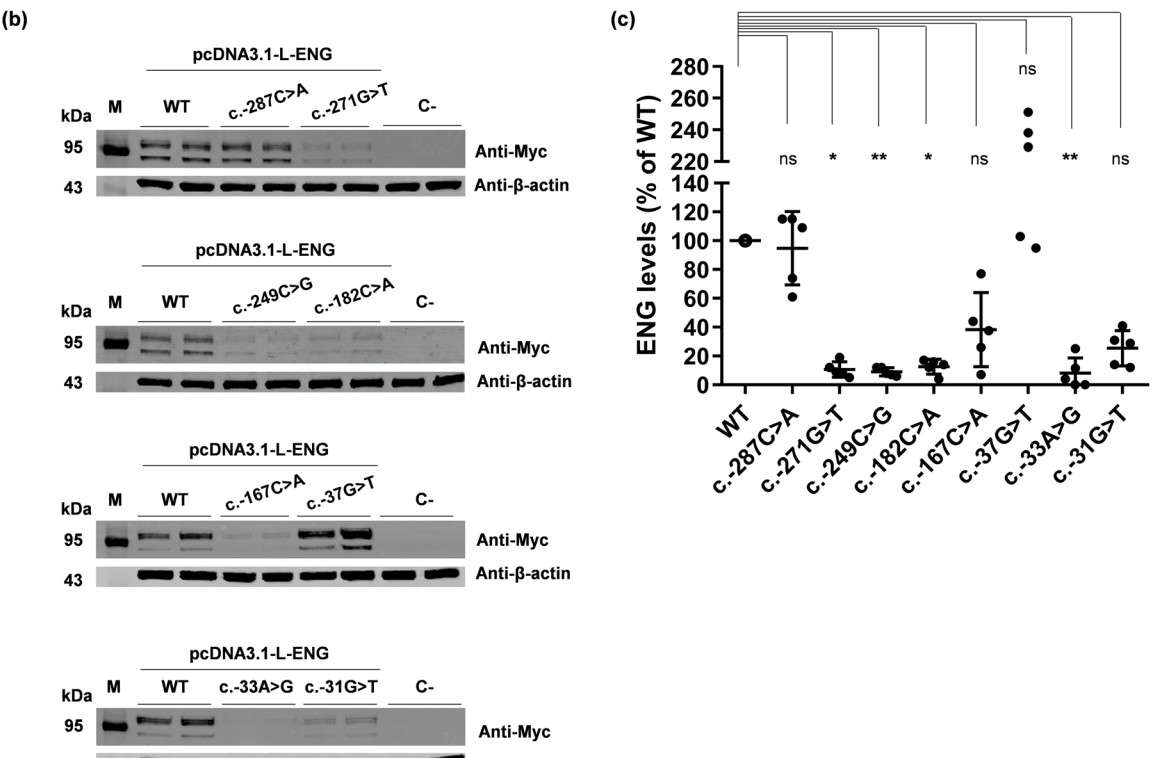

**Fig. 2 | *ENG* variants at the origin of uAUGs in frame with the same stop codon at position c.125. a** Eight variants in the 5'UTR create uAUG-initiated upstream overlapping open reading frames (uoORFs) with different sizes but ending at the same position. Position and identity of these variants and of created uAUG, size of the uoORFs, associated stop codon located at position c.125 and first and last positions of the CoDing Sequence (CDS) (c.1 and c.1977, respectively) are indicated. TSS, translation start site; CDS, CoDing Sequence. **b** Western blot results on total proteins extracted from transfected HeLa cells with 1 μg of pcDNA3.1-L-ENG constructs. Two bands of different molecular weights are observed for Endoglin likely corresponding to more glycosylated (upper band) and less/non glycosylated (lower band) ENG monomers. Anti-Myc corresponds to the used antibody for the target protein from HeLa and anti-β-actin corresponds to the antibody used against the reference protein. kDa, kilodalton; M, protein ladder; WT, wild type; C-, negative control corresponding to pcDNA3.1- empty vector. Shown results are representative of 5 independent experiments. **c** Quantification of Endoglin steady-state levels in HeLa cells from (**b**). For quantification, the average of each duplicate has been calculated from the quantified values and Endoglin levels for each sample have been normalized to the corresponding β-actin levels then to the WT (%). The two bands obtained for the Endoglin, corresponding to the more glycosylated (upper band) and less/non glycosylated (lower band) ENG monomers, were taken together for the quantification. Graphs with standard error of the mean are representative of five independent experiments. **, $p < 10^{-2}$; *, $p < 0.05$, ns, non-significant (Kruskal-Wallis followed by Dunn comparison test of variants versus WT).

surrounding a given canonical or non-canonical TIS to generate KSS scores reflecting the strength of Kozak sequences. The authors defined a threshold of 0.64 for translation initiation by uAUGs. We extracted KSS scores for our 13 uAUG-creating variants of interest (Table 1; Supplementary Data 2) and found that 6/13 uAUGs have scores higher than 0.64. These six variants (c.-271G > T, c.-249C > G, c.-142A > T, c.-127C > T, c.-79C > T and c.-33A > G) were observed to decrease the protein levels of Endoglin in our experiments. Interestingly, these variants are those showing the most drastic effects in our assays. However, 5/11 variants decreasing the protein levels (c.-182C > A, c.-167C > A, c.-68G > A, c.-31G > T and c.-10C > T) presented

**(a)**

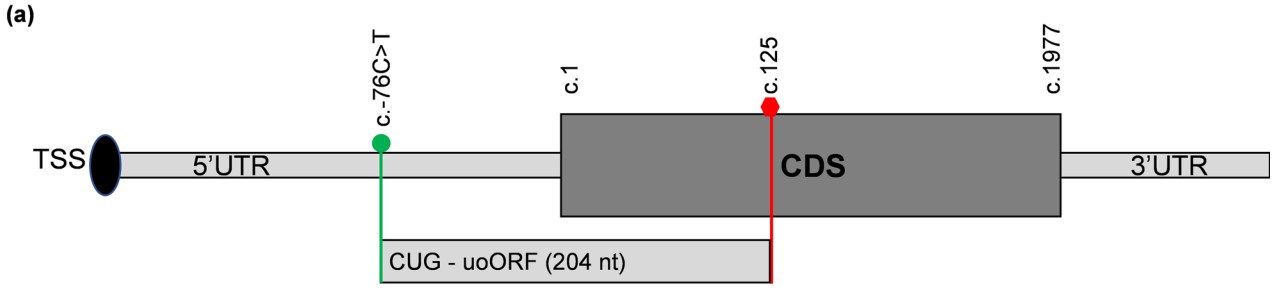

**(b)**

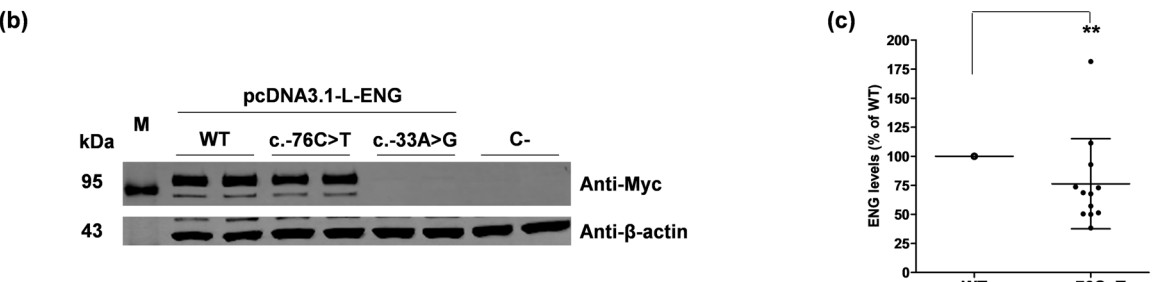

**(c)**

**Fig. 3 | uoORF-creating variants identified in French HHT patient. a** *ENG* c.-76C > T creating an upstream CUG in frame with the stop codon at position c.125 at the origin of an upstream Overlapping Open Reading Frame (uoORF) of 204 nucleotides (nt). Position and identity of the variant and of created uCUG, and first and last positions of the CoDing Sequence (CDS) (c.1 and c.1977, respectively) are indicated. **b** Western blot results on total proteins extracted from transfected HeLa cells with 1 μg of pcDNA3.1-L-ENG constructs. Two bands of different molecular weights are observed for Endoglin likely corresponding to more glycosylated (upper band) and less/non glycosylated (lower band) ENG monomers. Anti-Myc corresponds to the used antibody for the target protein from HeLa and anti-β-actin corresponds to the antibody used against the reference protein. kDa, kilodalton; M, protein ladder; WT, wild type; C-, negative control corresponding to pcDNA3.1-empty vector. Shown results are representative of five independent experiments. **c** Quantification of Endoglin steady-state levels in HeLa cells from (**b**). For quantification, the average of each duplicate has been calculated from the quantified values and Endoglin levels for c.-76C > T sample have been normalized to the corresponding β-actin levels then to the WT (%). The two bands obtained for the Endoglin, corresponding to the more glycosylated (upper band) and less/non glycosylated (lower band) ENG monomers, were taken together for the quantification. Graphs with standard error of the mean are representative of 12 independent experiments. **, $p < 10^{-2}$ (Kruskal-Wallis test of the c.-76C > T versus WT).

with KSS scores below the recommended threshold of 0.64 as the two variants with no decrease of protein levels.

Furthermore, we evaluated the efficiency of PreTIS scores to predict uTIS that can alter the protein levels. These scores predict the efficiency of a given uTIS to initiate the translation. We hypothesized that the highest PreTIS scores predicting the most efficient uTIS would be associated with the most drastic effect on the protein levels. We collected PreTIS scores for the 10/13 uAUGs created by 5'UTR variants in *ENG*. The remaining three variants being located within the first 99 nucleotides of the 5'UTR (c.-287C > A, c.-271G > T, and c.-249C > G), they cannot be predicted with PreTIS. The totality of the predicted uAUGs (10/10) has PreTIS scores higher than the predefined threshold for translation efficiency (0.54) (Table 1). Importantly, all but the c.-37G > T are associated with a decrease in the protein levels.

**New uoORF-creating variants identified in HHT patients**

We used the generated catalog of variants annotated with MORFEE to retrospectively analyze *ENG* variants detected in patients from the French National reference center for HHT with unresolved molecular diagnosis. We thus identified two uTIS-creating variants, the aforementioned c.-33A > G and the c.-76C > T variants. The first one, creating a uAUG-c.125 codon and never reported in public databases, was identified in a patient with definite HHT according to Curaçao criteria. Our experimental study (Fig. 2b-c) provided a strong argument for its pathogenicity. The second variant, creating a non-canonical TIS (uCUG), is also predicted to generate a uoORF ending at the c.125 codon (Fig. 3a). This variant (rs943786398) was detected in 2 unrelated patients referred for genetic screening for suspicious HHT, although they only fulfilled one of the Curaçao criteria (Supplementary Data 8). The proband in the first family had only a few

telangiectasias and recurrent epistaxis during childhood, with atypical presence of stroke and deep vein thrombosis. In the second family, the proband presented with pulmonary AVM, one epistaxis per month and not more than two telangiectasias on her face. In this family, the father was an asymptomatic carrier of the variant. This variant has been classified as a VUS associated to HHT in ClinVar. Following the same experimental workflow as above, we observed that the c.-76C > T variant was associated with decreased Endoglin levels of more than 25% in comparison with the WT (Fig. 3b, c; Supplementary Fig 4). Significant difference of *ENG* transcript amounts was observed between c.-76C > T and WT by RT-qPCR (Supplementary Fig. 3c), suggesting that additional potential effect of this variant on RNA should not be excluded. Very interestingly, the uCUG created by *ENG* c.-76C > T is encompassed by a strong Kozak sequence and carries very high scores with TIS-predictor and PreTIS (Table 1). Eight additional variants creating non-canonical uTIS in frame with the stop codon at position c.125 have been reported in ClinVar (Supplementary Data 4).

## Discussion

The diagnosis wandering in HHT is one of the main challenges for the molecular diagnosis and management of HHT patients and their families. This could be explained by the lack of information about non-coding or structural variants in HHT genes. Recent studies have, however, shown the association of this kind of variants in HHT patients, suggesting that they are still underexplored[6,7]. Curiously, at least five non-coding variants carrying very similar characteristics in the 5'UTR of *ENG* have been associated with HHT[8-13]. They all create overlapping upORFs ending with the same stop codon at position c.125 and are associated with severe HHT phenotypes[13], being a very particular characteristic for *ENG*.

To tackle the challenge of diagnostic wandering, we conducted an exhaustive characterization of 5'UTR variants altering upORFs in *ENG*. We started by bioinformatically characterizing all 5'UTR *ENG* SNVs that can create or modify upORFs, aiming to facilitate the identification of pathogenic variants for HHT. This was achieved by cataloging all possible SNVs (already reported or yet unreported ones) that can create canonical or non-canonical uTIS, create new upstream stop codons or delete existing stop codons in the 5'UTR of *ENG*. From this catalog, we were able to identify, in patients followed by the French National Reference Center for HHT, two extremely rare variants including one, c.-33A > G, that had never been reported and another, c.-76C > T, classified as VUS in ClinVar. Both of these variants create uoORFs ending with stop codon located at c.125 position and are associated with a decrease of Endoglin levels in our experimental assays. While the c.-33A > G is predicted to create a uAUG, the c.-76C > T creates a uCUG and is the first example of 5'UTR creating a non-canonical TIS in HHT. This variant was identified in 2 patients with unlikely HHT. These findings suggest that rare *ENG* variants predicted to create non-canonical uTIS in frame with the uStop-c.125 should also be considered as candidates for causing HHT or explaining clinical heterogeneity. Moreover, we demonstrated that most uAUG-creating variants at the origin of uoORFs ending with the uStop-c.125 are associated with decreased Endoglin levels in vitro. It would be interesting to conduct similar experimental investigations for variants creating uTIS in frame with the uStop at position c.90 (Fig. 2c), thus generating similar type of upORFs (uoORFs), and to better explore the context of these stop codons that could influence the effects of the associated uoORFs.

Beyond the 2 identified variants in French HHT patients, the generated bioinformatics catalog (Supplementary Data 2), together with our experimental findings will serve to identify candidate rare 5'UTR *ENG* variants in HHT, contributing to resolve molecular origins of HHT. For instance, among *ENG* variants reported in ClinVar, 56 are located in the 5'UTR with 18 ( ~ 42% of 5'UTR variants) annotated with MORFEE as creating uTIS. Very importantly, the recent availability of a comprehensive catalog of all 5'UTR variants with the potential to alter upORFs across all human genes[23] will greatly facilitate the identification of novel 5'UTR pathogenic variants in other HHT genes and beyond. Of note, we also showed that prediction metrics dedicated to upORF-altering variants (Kozak strength, KSS and PreTIS scores) deserve to be evaluated with a larger dataset of variants with different impacts (increase, decrease or null effect) on protein levels, in particular for variants with predicted multiple consequences on upORF alteration (see Supplementary Data 3 on c-37C > T).

In conclusion, our work provides new insights into the interpretation of *ENG* non-coding variants in HHT patients.

## Methods

### Nomenclature
DNA sequence variant nomenclature follows current recommendations of the HGVS[27].

### Search for all upORF-altering variants in the 5'UTR of *ENG*
We in silico mutated each position in the 5'UTR of the main transcript of *ENG* (MANE select, ENST00000373203.9) reported in the latest version of Ensembl database (GRCh38.p14) with the three alternative nucleotides to generate a vcf file containing all possible SNVs between positions c.-303 (i.e., first nucleotide of the 5'UTR) and c.-1 (Supplementary Data 1). The generated vcf file was then annotated using an updated version of the MORFEE bioinformatics tool[23] now available on https://doi.org/10.5281/zenodo.14864790. This version can annotate variations predicted to (i) create canonical and non-canonical TIS; (ii) create new stop codons (TAA, TAG and TGA); and/or (iii) delete existing stop codons along a given transcript. The resulting list of SNVs is provided in Supplementary Data 2.

Additionally, we extracted known upORFs in the 5'UTR of *ENG* from public databases reporting small ORFs that have been identified through ribosome profiling and/or mass spectrometry in human cells: sORFs repository (www.sORFs.org)[24]; metamORF database (https://metamorf.hb.

univ-amu.fr/)[25]; vUTR interface (https://vutr.rarediseasegenomics.org/); and smoRFs browser (https://smorfs.ddnetbio.com)[26].

### Selection of *ENG* variants for experimental validation
Five rare variants creating uAUGs in frame with the stop codon at position c.125 have been previously identified in HHT patients and have shown drastic effects on Endoglin protein levels[13]. We here selected all additional eight variations identified by MORFEE to create uAUGs in frame with the stop codon for experimental validation (Supplementary Data 2).

We also selected the c.-76C > T variation, creating a non-canonical uTIS in frame with the c.125 stop codon. This variation was further identified in a collection of HHT patients with unresolved molecular diagnosis (Supplementary Data 2).

### Plasmid constructs
Preparation of pcDNA3.1-L-ENG (L, Long, NM_001114753.3) constructions was performed by directed mutagenesis on pcDNA3.1-L-ENG-WT construct[13] using the 2-step overlap extension PCR method[28] and primers listed in Supplementary Data 9. *BamHI* and *SacII* or *BamHI* and *BlpI* were used as cloning sites, depending on the inserted variant (Supplementary Data 9). Only pcDNA3.1-L-ENG-c.-287C > A construction was prepared by a simple PCR reaction with a forward primer carrying the variant downstream of *BamHI* cloning site and a reverse primer overlapping *SacII* (Supplementary Data 9). All new constructs were verified by Sanger sequencing of the insert and cloning sites (Azenta/Genewiz).

### Functional analysis of *ENG* 5'UTR variants
Transfection of HeLa cells, RNA and protein extractions as well as western blot and RT-qPCR analyses were carried out as described in Soukarieh et al., 2023[13]. Briefly, the generated pcDNA3.1-L-ENG constructs were transfected in HeLa cells (ATCC) cultured in RPMI medium (Gibco-Invitrogen) supplemented with 10% fetal calf serum (Gibco-Invitrogen). WT, mutant, and empty vector (negative control) constructs were transfected in parallel and with technical duplicates for each construct in 6-well plates at cell confluence of 60-80% by using JetPRIME® reagent (Polyplus Transfection) according to the manufacturer's recommendations. Total RNA and protein were extracted from the same well by using the RNeasy mini kit (Qiagen) and RIPA buffer supplemented with proteases, respectively, 48 h after the transfection.

Same quantities of proteins were analyzed on 10% SDS-PAGE gels after measurement with the BCA protein assay kit (Pierce™). Protein transfer was performed by using the trans-blot turbo transfer system on PVDF membranes (Bio-Rad). Anti-(c-Myc Tag) antibody (Merck Millipore, #05–419) and anti-β-actin (Cell Signaling, #4970) were used at 1/3000 dilutions and were targeted with fluorescent goat anti-mouse IgG Alexa Fluor 700 (ThermoFisher, #A-21036, 1/5000) goat anti-Rabbit IgG (H + L) Alexa Fluor 750 (Invitrogen, #A-21039, 1/5000), respectively. Blots were scanned by using the Odyssey Infrared Imaging System (Li-Cor Biosciences) in 700 and 750 channels. Quantification was performed by taking into account the two bands obtained for the Endoglin and corresponding to the more glycosylated (upper band) and less/non glycosylated (lower band) ENG monomers[5]. For each construct, average of technical duplicates was calculated then normalized to the corresponding β-actin values, before normalizing to the WT.

Total RNA was analyzed by reverse transcription (M-MLV reverse transcriptase from Promega) using oligo-dT and random hexamers followed by a qPCR reaction (duplicate/sample) on 20 ng of cDNA with ENG- or α-tubulin-specific primers (Supplementary Data 9; reaction efficiency between 90–110%) and the GoTaq qPCR Master mix (Promega), in the presence of CXR reference dye. qPCR reactions were ran on QuantStudio3 Real-Time PCR System (ThermoFisher) with technical duplicates for each sample. Results were analyzed with the QuantStudio design and analysis software. Relative amounts of ENG to α-Tubulin were calculated by using the $2 - \Delta\Delta CT$ method.

## Kozak sequence interpretation and bioinformatics predictions

We here used the obtained results in our functional assays on *ENG* variants (Table 1) to evaluate the predictive power of the strength of the Kozak sequence and of two predictive scores, TIS-predictor[29] (KSS scores) and PreTIS[30] tools.

The Kozak sequence is defined as the genomic sequence surrounding a TIS. The optimal Kozak sequence is [**A/G**]CC<u>ATG</u>**G**, underlined nucleotides corresponding to the TIS and bolded nucleotides to the most conserved positions. We have arbitrarily considered a given Kozak sequence as (i) strong when it contains a purine at position −3 <u>and</u> a guanine at position +4; (ii) moderate when it contains a purine at position −3 <u>or</u> a guanine at position +4, and; (iii) weak when it does not contain a purine at position −3 nor a guanine at position +4.

## Statistics and reproducibility

Differential protein and RNA levels were assessed using Kruskal-Wallis test followed by Dunn test for handling multiple comparison when appropriate. A threshold of $p < 0.05$ was used to declare statistical significance.

Every experiment was repeated at least five times with technical replicates for each sample. For each experiment, points on graphs represent the means of replicates from the same sample. The precise number of repeated experiments is indicated in each figure.

## Reporting summary

Further information on research design is available in the Nature Portfolio Reporting Summary linked to this article.

## Data availability

ENG constructs generated during this study are available upon request by email from the corresponding author (omar.soukarieh@inserm.fr). All details about *ENG* 5'UTR SNVs analyzed with MORFEE, clinical data for HHT patients carrying *ENG* uoORF-creating variants, primers' sequence and identity, as well as raw experimental data analyzed in this work are provided in Supplementary Data 1-9.

## Code availability

The used version of MORFEE tool is available at https://doi.org/10.5281/zenodo.14864790.

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

## Acknowledgements
This project was carried out in the framework of the French National Research Agency (ANR) ANR-23-CE17-0042-01 program as part of the ENDOMORF project, of the INSERM GOLD Cross-Cutting program (D-A.T.) and of the Precision and global vascular brain health institute funded by the France 2030 investment plan as part of the IHU3 initiative under grant agreement ANR-23-IAHU-0001. O.S was financially supported by a grant of the Lefoulon-Delalande Foundation.

## Author contributions
O.S. and D.A.T. conceived the project. O.S., C.P. and B.J.V. designed the experiments. O.S., C.P., C.D. and B.J.V. performed the experiments. O.S., C.P., C.D., and B.J.V. analyzed the data. B.J.V. and A.G. provided technical support and suggestions on the project and the experiments. S.M., A.G.u., S.D.G. and M.T. were in charge of clinical management of HHT patients. C.M. performed the mutational saturation and variant annotation with MORFEE. O.S. and D.A.T. drafted the paper that was further shared with co-authors who read/corrected/ and approved the final manuscript.

## Competing interests
The authors declare that they have no known competing financial or non-financial interests or personal relationships that could have appeared to influence the work reported in this paper.
