## [Transparent Peer Review file · Communications Biology]

Overlapping upstream ORFs ending at c.125 lead to reduced Endoglin, contributing to Hereditary Hemorrhagic Telangiectasia

Corresponding Author: Mr Omar Soukarieh

Version 0:

Reviewer comments:

Reviewer #1

(Remarks to the Author)

Focusing on just one of the two original approaches enhances the clarity and impact of the message. The genetic study is well-executed and informative, providing valuable insights into the role of ENG uORFs ending at c.125 in the molecular genetics of HHT. The mechanistic investigation of the identified variants, previously considered somewhat superficial in earlier submissions, is now more focused and exclusively addresses ENG expression in cell culture systems. While further work will be necessary to conclusively demonstrate that these identified variations are functional and thus of diagnostic value, the present work offers important new insights into the molecular origin of the undiagnosed cases of HHT.

Reviewer #2

(Remarks to the Author)

Does the manuscript have technical or conceptual flaws that should prohibit its publication? If so, please provide details.

The manuscript is complete technically, and conceptually well based and experimentally demonstrated by in silico, in vitro and by cases found in the HHT database of French patients.

Are the conclusions original? If not, please provide relevant references.

The conclusions are original since authors open new sources of variants which are pathogenic to explain unsolved cases of clinically diagnosed HHT who did not have mutational cause. Some references in the same or similar field are mentioned in the manuscript. These references deal with some of the variants they present in a complete study of all the variants. Also they mention references applicable not only to the case of variations in Endoglin and HHT, but for other diseases caused by mutations, where so far no genetic pathogenic variants were found.

Do you feel that the results presented are of immediate relevance for people in your own discipline or for a broader audience? If you recommend publication, please outline briefly what you consider to be the outstanding features.

Undoubtedly the results are of immediate relevance working in HHT rare disease to find the clinical variant causing the disease. Any researcher in the field of HHT with genetically unsolved cases, may start already exploring the 5' upstream Endoglin promoter following the findings of this manuscript. But as the authors state in the discussion, the conclusions are not only valid for HHT, but for any other hereditary disease genetically unsolved.

If you feel that specific additional experiments would strengthen the case for publication in Communications Biology, please provide suggestions.

I think that the authors cover all the possibilities to test their hypothesis: in silico saturation of the variants giving rise to uoORF or UTIS in the upstream region around the main transcription initiation site for Endoglin. Authors look for cases of these variants in the HHT french database and also test in vitro some of the variants to check their consequences on RNA and protein levels.

REVIEWERS' COMMENTS:

Please find in blue, our responses to the reviewers.

Reviewer #1 (Remarks to the Author):

Focusing on just one of the two original approaches enhances the clarity and impact of the message. The genetic study is well-executed and informative, providing valuable insights into the role of ENG uORFs ending at c.125 in the molecular genetics of HHT. The mechanistic investigation of the identified variants, previously considered somewhat superficial in earlier submissions, is now more focused and exclusively addresses ENG expression in cell culture systems. While further work will be necessary to conclusively demonstrate that these identified variations are functional and thus of diagnostic value, the present work offers important new insights into the molecular origin of the undiagnosed cases of HHT.

We thank the reviewer for the very positive feedback. We believe that by making our results publicly available, we increase the chances of defining the diagnostic values of the studied variants. Moreover, we currently develop additional assays to better characterize their functional effect.

Reviewer #2 (Remarks to the Author):

Does the manuscript have technical or conceptual flaws that should prohibit its publication? If so, please provide details.

The manuscript is complete technically, and conceptually well based and experimentally demonstrated by in silico, in vitro and by cases found in the HHT database of French patients.

We are glad to read this.

Are the conclusions original? If not, please provide relevant references.

The conclusions are original since authors open new sources of variants which are pathogenic to explain unsolved cases of clinically diagnosed HHT who did not have mutational cause. Some references in the same or similar field are mentioned in the manuscript. These references deal with some of the variants they present in a complete study of all the variants. Also they mention references applicable not only to the case of variations in Endoglin and HHT, but for other diseases caused by mutations, where so far no genetic pathogenic variants were found.

We thank the reviewer for the positive comment.

Do you feel that the results presented are of immediate relevance for people in your own discipline or for a broader audience? If you recommend publication, please outline briefly what you consider to

be the outstanding features.

Undoubtedly the results are of immediate relevance working in HHT rare disease to find the clinical variant causing the disease. Any researcher in the field of HHT with genetically unsolved cases, may start already exploring the 5' upstream Endoglin promoter following the findings of this manuscript. But as the authors state in the discussion, the conclusions are not only valid for HHT, but for any other hereditary disease genetically unsolved.

We agree that the 5'UTR of all transcripts implicated in human diseases should now be largely explored.

If you feel that specific additional experiments would strengthen the case for publication in Communications Biology, please provide suggestions.

I think that the authors cover all the possibilities to test their hypothesis: in silico saturation of the variants giving rise to uORF or UTIS in the upstream region around the main transcription initiation site for Endoglin. Authors look for cases of these variants in the HHT french database and also test in vitro some of the variants to check their consequences on RNA and protein levels.

We thank the reviewer and hope that this study will contribute to resolve more unexplained HHT cases.